# Comparative Study of Cytokine Storm Treatment in Patients with COVID-19 Pneumonia Using Immunomodulators

**DOI:** 10.3390/jcm11102945

**Published:** 2022-05-23

**Authors:** Felicia Marc, Corina Maria Moldovan, Anica Hoza, Sorina Magheru, Gabriela Ciavoi, Dorina Maria Farcas, Liliana Sachelarie, Gabriela Calin, Laura Romila, Daniela Damir, Alexandru Gratian Naum

**Affiliations:** 1Clinical Departament, Faculty of Medicine and Pharmacy, Oradea University, University Street 1, 410087 Oradea, Romania; feliciamarc.dr@gmail.com (F.M.); cmold2003@yahoo.com (C.M.M.); anica_hoza@yahoo.com (A.H.); sorinamagheru@yaoo.com (S.M.); gabrielaciavoi@yahoo.com (G.C.); dmfarcas@yahoo.com (D.M.F.); 2Preclinical Department, Apollonia University, Păcurari Street 11, 700511 Iași, Romania; m_gabriela2004@yahoo.com; 3Department of Medical Disciplines, Grigore T. Popa University of Medicine and Pharmacy Iasi, University Street 16, 700115 Iasi, Romania; alexandru-gratian.naum@umfiasi.ro

**Keywords:** cytokine storm, immunomodulators, biochemical parameters

## Abstract

(1) Background: In patients hospitalized with COVID-19 pneumonia, especially moderate and severe forms, a cytokine storm may occur, characterized by the worsening of symptoms and the alteration of biological parameters on days 8–12 of the disease. The therapeutic options for cytokine storms are still controversial, requiring further clarification; (2) Methods: Our study included 344 patients with moderate and severe pneumonia admitted to the internal medicine department who developed a cytokine storm (diagnosed by clinical and biochemical criteria). In group A, 149 patients were treated with Remdesivir and Tocilizumab (together with other drugs, including corticosteroids, antibiotics and anticoagulants), and in group B, 195 patients received Remdesivir and Anakinra. Patients were monitored clinically and by laboratory tests, with the main biochemical parameters being CRP (C-reactive protein), LDH (lactic dehydrogenase) and ferritin; (3) Results: Patients were followed up from a clinical point of view and also by the measurement of CRP, LDH and ferritin at the beginning of therapy, on days three to four and on the tenth day. In both groups, we registered a clinical improvement and a decrease in the parameters of the cytokine storm. In group A, with the IL-6 antagonist Tocilizumab, the beneficial effect occurred faster; in group B, with the IL-1 antagonist Anakinra, the beneficial effect was slower. (4) Conclusions: The use of the immunomodulators, Tocilizumab and Anakinra, in the cytokine storm showed favorable effects, both clinical and biochemical.

## 1. Introduction

During the pandemic caused by the SARS-CoV-2 virus, many hospitals became COVID-19 support hospitals. The most commonly affected organ in this disease is the lung; thus, the virus may cause viral pneumonia, which may be classified following a clinical and paraclinical evaluation as mild, moderate or severe. Usually, a superimposed bacterial infection occurs, so pneumonia becomes viral and bacterial [1,2,3,4].

In addition to lung damage (cough, shortness of breath, chest pain, hemoptysis, wheezing, rhinorrhea, odynophagia, nasal congestion, anosmia and ageusia), the SARS-CoV-2 virus may cause other organs and systems involvement, such as general manifestations (fever, chills, fatigue myalgias, arthralgias and hydro electrolytic disorders), neurological manifestations (headache, vertigo, polyneuropathy, Guillain–Barre syndrome, stroke, encephalitis and meningitis), psychiatric manifestations (confusion, disorientation, psychomotor agitation, depression, anxiety and xenophobia), cardiovascular manifestations (myocarditis, pericarditis, heart failure, arrhythmias and myocardial infarction), thromboembolic manifestations (pulmonary embolism, deep vein thrombosis and disseminated intravascular coagulation), endocrine manifestations (diabetes that may newly onset, or an aggravated one, adrenal insufficiency and subacute thyroiditis), gastroenterological manifestations (hepatic impairment, nausea, vomiting, abdominal pain and diarrhea) and renal impairment (hematuria and renal failure) [5,6,7].

In COVID-19 pneumonia, on days 8–12 of the disease, some patients develop the ‘‘cytokine storm’’. From a clinical point of view, patients have fever, fatigue, anorexia, headache, diarrhea, dyspnea, cough, cyanosis, and require oxygen therapy [8,9,10]. 

The laboratory findings include elevated CRP (C-reactive protein) and D-dimers, leukocytosis or leukopenia, anemia, thrombocytopenia and prominent levels of serum inflammatory cytokine levels, Interferon-gamma, IL-6, IL-1 and IL-10.

There is a consensus that the biochemical criteria for the detection of the cytokine storm are: the elevation of C-reactive protein (CRP) over 30 mg/L (normal value 0–5 mg/L), the increase of lactate dehydrogenase (LDH) over 300 u/L (normal value 0–247 u/L) and the increase of ferritin over 700 ng/mL (normal value 10–291 ng/mL) [11,12].

If the cytokine storm is left untreated, or the treatment is unsuccessful, tachypnea and hypoxemia may lead to ARDS, acute respiratory distress syndrome, or other complications like acute kidney injury, liver damage, stress-related cardiomyopathy and disseminated intravascular coagulopathy with bleeding risk and death.

Studies show that COVID-19 patients have increased levels of numerous inflammatory cytokines, like IL-1 beta, IL-2, IL-6, TNF-alpha, etc. and these cytokines correlate with disease severity. The release of numerous cytokines is associated with clinical manifestations like headache, chills, dizziness, fatigue, fever, lung damage, vascular leaking, heart failure and the synthesis of acute-phase proteins [13]. 

Targeting the cytokine storm was done in practice with different medications designed to treat other diseases. Cytokine inhibitors, like the IL-1 antagonist Anakinra or the IL-6 antagonist Tocilizumab, were used for immuno-rheumatological pathologies, and their use in COVID-19 patients requires validation.

### Aim

This study aimed to follow the response to the treatment of patients with COVID-19 viral pneumonia in moderate and severe forms in which a cytokine storm occurs and develops [13,14,15].

The classic treatment plan, according to the Romanian Ministry of Health recommendations protocol, includes an antiviral (Favipiravir or Remdesivir), anticoagulant, corticosteroids (Dexamethasone, which acts as an anti-inflammatory and an immunomodulator), antibiotic and symptomatic (Acetaminophen for fever, Codeine phosphate or Acetylcysteine for cough, Metoclopramide for vomiting, etc.) [16,17,18]. Compared to the classic plan mentioned above, in severe cases, or those with cytokine storms, the treatment can be adjusted: Favipiravir is replaced with Remdesivir, and immunomodulators are added, Tocilizumab (IL-6 antagonist) (Roactemra) or Anakinra (IL-1 antagonist) (Kineret). Treatment options includedRemdesivir with Tocilizumab and Remdesivir with Anakinra, along with corticosteroids (in the cytokine storm, Dexamethasone was given in progressively increasing doses), antibiotics and anticoagulants [19,20,21].

## 2. Materials and Methods

The present study followed up patients hospitalized in the Oradea Municipal Clinical Hospital, Romania, a COVID-19 support hospital, within the internal medicine department between November 2020 and November 2021. The patients included in the study were patients diagnosed with SARS-CoV-2 viral infection by the rapid antigen test or by collection of nasopharyngeal exudates using the RT-PCR method. 

Patients of different ages, women and men, from rural and urban areas, with various associated comorbidities, with various forms of viral pneumonia, were evaluated. The patients underwent an anamnesis, which included the symptoms that led to hospitalization, hereditary history, personal physiological and pathological history, home treatments, living and working conditions, and then the objective examination was performed. Patients were investigated both clinically and paraclinically, but also in terms of treatment. From a clinical point of view, the general condition (asthenia, adynamia, fever, loss of appetite and myalgia), oxygen saturation (measured with a pulse oximeter), temperature, blood pressure and pulse were monitored.

From the paraclinical point of view, regarding the laboratory analysis, attention was focused on the values of the parameters such as ESR (erythrocyte sedimentation rate), CRP (C-reactive protein), full blood count, blood glucose, LDH (lactic dehydrogenase), CK (creatine kinase), GOT (aspartate aminotransferase), GTP (alanine aminotransferase), procalcitonin, D-dimers, ferritin and ionogram.

All patients underwent chest computer tomography, native or contrast-enhanced, to assess the presence and severity of viral pneumonia (possibly associated with bacterial superinfection), the presence of pulmonary embolism and other organ lesions caused by the coronavirus (head, abdominal, pelvic CT, etc.).

After evaluation and classification, the treatment was administered according to the protocols of the Ministry of Health from Romania in force at that time (there were three during the period). The treatment regimen included an antiviral (Favipiravir or Remdesivir), an antibiotic, anticoagulant, corticosteroids and, if necessary, immunomodulatory supplementation with Tocilizumab or Anakinra [22,23,24].

The patients with moderate to severe forms of viral pneumonia who had developed a cytokine storm (worsening of symptoms on days 8 to 10 of the disease, fever, dyspnea with tachypnea, decreased oxygen saturation and increased three parameters, including CRP, LDH and ferritin) were classified in two groups of patients: group A—patients who received treatment with Remdesivir and Tocilizumab, and group B—patients who received treatment with Remdesivir and Anakinra.

In each group, we monitored the inflammatory parameters (CRP, ESR and Fibrinogen, monitoring especially the CRP value during hospitalization), as well as the parameters of disease severity, including lymphopenia, thrombocytopenia, LDH, CK (creatine phosphokinase) and ferritin (summarizing LDH and ferritin) in patients who developed cytokine storms.

Regarding the procoagulant status, we followed D-dimers as the highlighting parameter. Several patients had bacterial superinfection, evidenced by an increase in procalcitonin. Patients with procalcitonin above 1.5 (the normal values being 0–0.3 ng/mL) were not included in the immunomodulatory therapy. Data were processed using SPSS 20 program. Statistical significance tests were performed by the χ^2^ method and ANOVA (Brown–Forsythe) was used to compare the means.

## 3. Results

From 1071 patients diagnosed and treated in the internal medicine department, there were recorded 134 deaths (after the transfer to the intensive care unit, many of them developed a cytokine storm, pulmonary thromboembolism and multiple organ failure). From the 937 patients treated and discharged, 344 developed cytokine storms, representing 36.39% of patients. The distribution by age groups, sex and rural/urban background was as follows:

### 3.1. Characteristics of the Population (Sex, Age and Environment)

In our study group, over 50% were women (59.10%), with the ratio of women/men being1:1, the age was between 26–88 years, with the average age as 61.18 years, and the patients came mainly from the urban environment (56.18%), with the urban/rural ratio being 1.3:1 (Table 1).

### 3.2. The Severity of the Disease

The classification of the disease by mild/moderate/severe pneumonia is based on the symptomatology, the biological investigations and the thoracic CT examinations, as observed in Table 2. 

Patients who developed cytokine storms during hospitalization were among those classified as moderate and severe forms of the disease; thus, among the moderate forms, 146 patients developed cytokine storms, and among the severe forms of the disease, 198 patients developed cytokine storms.

In the Remdesivir and Anakinra group, women predominated (60.92%), with the percent being significantly higher than in the Remdesivir and Tocilizumab group (39.08%, *p* = 0.102), (Table 3).

Group A included 149 patients with cytokine storms that were treated with Remdesivir and Tocilizumab. Among the patients included in group A, 125 had comorbidities (hypertension, diabetes, obesity, neoplastic disease and liver disease), representing 84.45%, (Table 4). 

According to the above table, we observed that obesity was the condition that correlated most with the worsening of symptoms and the development of cytokine storms. Obese patients also developed most cases of pulmonary thromboembolism. Patients who developed a cytokine storm from group A had the following comorbidities: obesity (34%), diabetes mellitus (22%), hypertension (16%), chronic obstructive lung disease (13%), liver diseases (9%) and malignancies (6%). Usually, patients had one or more comorbidities, such obesity and diabetes or diabetes and hypertension.

Group B included 195 patients with cytokine storms treated with Remdesivir and Anakinra. Among the patients in group B, 169 had comorbidities, representing 87.56%.

We registered that those patients who developed cytokine storms from group B had at least one comorbidity: obesity (30%), diabetes mellitus (27%), hypertension (15%), chronic obstructive lung disease (10%), malignancies (10%) and liver diseases (8%). All of the patients had a combination of two or more comorbidities.

Thus, for group A (cytokine storm patients treated with Remdesivir and Tocilizumab):

### 3.3. Evaluation of CRP (Normal Values 0–5 mg/L)

The mean value for CRP was 132.2 mg/dL on day 1 of treatment, on days 3–4 it was 85 mg/dL and on day 10 it reached 24.6 mg/dL. Patients remained in the hospital until a clinical and biological improvement (return to normal parameters) that allowed discharge. When discharged, some patients needed oxygen therapy at home (Figure 1).

### 3.4. Evaluation of Lacticodehydrogenase (Normal Values 0–247 u/L, in Cytokine Storm, Increases over 300 u/L)

The average LDH value at the start of treatment was 475.6 u/L, on days 3–4 it was 327.2 u/L and on day 10 it was 221.1 u/L, thus falling back to normal values (Figure 2).

### 3.5. Evaluation of Ferritin (Normal Values 10–291 ng/mL, in Cytokine Storm, Increases to over 700 ng/mL)

The average value of ferritin in day 1 of the treatment was 860.1 ng/mL, on day 3–4 it was 721.0 ng/mL and on day 10 it reached 302.8 ng/mL, very close to normal values (Figure 3).

Group B, which included cytokine storm patients treated with Remdesivir and Anakinra:

### 3.6. Evaluation of CRP Variation (Normal Values 0–5 mg/L)

The mean value of CRP in day 1 of the treatment was 130.3 mg/L, on day 3–4 it was 108.1 mg/L and on day 10 it reached 40.9 mg/L (Figure 4).

### 3.7. The Evaluation of Lacticodehydrogenase (Normal Values 0–247 u/L, in Cytokine Storm, Increasesto over 300 u/L

The average LDH value on day 1 of the of treatment was 475.4 u/L, on days 3–4 it was 403.2 u/L and on day 10 it was 252.8 u/L, approaching normal values (Figure 5).

### 3.8. Evaluation of Ferritin (Normal Values 10–291 ng/mL, in Cytokine Storm, Increases to over 700 ng/mL)

The average value of ferritin on day 1 of the treatment was 861.5 ng/mL, on day 3–4 it was 743.0 ng/mL and on day 10 it reached 318 ng/mL, less close to normal compared to group A (Figure 6).

#### 3.8.1. Comparative Study

##### The Evaluation of CRP

If the evolution of CRP between the two groups is compared, the initial value at which the treatment for the cytokine storm started is approximately similar, 132 vs. 130.3 mg/dL. Subsequently, on days 3–4, the CRP value decreased more in group A (85 vs. 108 mg/dL), and on the 10th day of treatment, the CRP value in group A reached 24.6 mg/dL, while the CRP value in group B was 40 mg/dL (<0.001). The decrease in CRP was faster and was maintained until day 10 in group A, as shown in Table 5. This was clinically correlated with a faster improvement in symptoms.

##### The Evaluation of LDH

The comparative evolution of LDH in the two groups shows an almost equal initial value (475.6 vs. 475.4 u/L), on days three to four, a more pronounced decrease is observed in group A (327.2 vs. 403.2 u/L), and on the tenth day, the LDH value is lower in group A, reaching normal values (221.1 vs. 252.8 u/L in group B), as shown in Table 6. We noticed that patients from group B, treated with Anakinra, had almost the same values at discharge as they were on the 10th day (254 u/L at discharge, 252 u/L on the 10th day); the clinical status was not aggravated, and this finding may be correlated with the slower evolution of patients from group B compared with group A.

The comparative analysis of the ferritin value between group A and group B shows an almost similar value at the initiation of treatment (860 vs. 861.5 ng/mL). Subsequently, on days 3–4, the ferritin decreases more in group A (721 vs. 743 ng/mL), and the difference is maintained untilday 10, when the ferritin in group A was 302 ng/mL, and in group B it was 318 ng/mL. On the 10th day, ferritin was still above normal (291 ng/L) in both batches, as shown in Table 7. The decrease in the ferritin value was slower, reaching normal values through days 13–14 of monitoring.

## 4. Discussion

In COVID-19-hospitalized patients, it is important to a clinical and biological evaluation to detect the appearance of the cytokine storm so it can be treated as early as possible [23,24,25,26]. A recent article by Christian Zanza et al. reviewed the therapeutic options for the cytokine storm in COVID-19, showing that the situation is still unclear and controversial [27,28].

Thus, for Tocilizumab, the COV-ACTA-Trial, REMAP-CAP and RECOVERY studies have shown that the IL-6 inhibitor is effective in patients with hypoxia and hyperinflammatory status, improving several clinical parameters, including mortality at 28 days [29,30].

In our study, the combination of Remdesivir and Tocilizumab was effective in moderate and severe forms of the disease, improving the clinical condition, oxygen saturation and the three main parameters monitored, including C-reactive protein, lactic dehydrogenase and ferritin. On day 10 of the treatment, LDH returned to normal, with CRP and ferritin very close to normal.

It should be mentioned that the monitoring of patients was performed both clinically (improvement of general condition, decrease in fever, tachypnea and increase in oxygen saturation) and biochemically (both cytokine storm parameters, but also hemoleukogram, procalcitonin, D-dimers, etc.), and through medical imaging, with patients remaining in the hospital for as long as necessarybeyond the 10th day.

Regarding Anakinra in the treatment of the cytokine storm in COVID-19, the results are still uncertain [28]. While some cohort studies have shown a clear benefit from the use of the IL-1 antagonist in severely ill patients, a recent RCT study in France was stopped due to the lack of usefulness of Anakinra, being observed that the treated patients had no improvement (it seems that the maximum benefit would be for those with severe forms, with high CRP values).

The combination of Remdesivir and Anakinra had a favorable effect, both in the moderate and severe forms that developed cytokine storms, improving the clinical condition, oxygen saturation, as well as CRP, LDH and ferritin. On the 10th day of treatment, all three parameters approached normal values. 

The hyperinflammatory status which occurs in a cytokine storm is caused by the release of numerous proinflammatory cytokines, such as IL-6, IL-1, TNF-alpha, etc., with IL-6 as the orchestral lead for COVID-19 cytokine release syndrome [30,31]. The secretion of IL-6 causes vascular leak syndrome, triggering coagulation and complement pathways; moreover, IL-6 plays a crucial role to induce a panoply of other proinflammatory cytokines and chemokines, continuing the process of inflammation and thrombosis that can finally produce multiple organ failure [31,32].

Tocilizumab is currently standing out as an IL-6 receptor blockade that might interrupt the inflammatory cascade at a crucial stage. Results from REMAP-CAP study suggest that Tocilizumab is most effective when administered early (within 24 h of organ failure) in patients with progressive disease and substantial oxygen requirements.

The primary laboratory indicator that determines the proper time to start Tocilizumab should be the blood IL-6 level; similarly, in order to start Anakinra, the levels of IL-1 should be determined. Unfortunately, these tests were not available for us.

The dose of Tocilizumab is usually a single dose of 800 mg, given as a single infusion. Anakinra is given for 7–10 days, subcutaneously or intravenously, at 1–2 ampoules of 100 mg.

The clinical effect is faster and more spectacular with Tocilizumab, and the clinical and biochemical parameters improve faster. Thus, in group A, CRP decreases more than in group B on day 3–4, and at day 10, a difference between CRP is maintained in the two groups, with the group B CRP value being higher. The same aspect is observed regarding the other two parameters, LDH and ferritin, with the decrease being higher on days 3–4 in the case of group A and being maintained later. The slowest decrease was observed, in both groups, for ferritin.

The aim of our study was to observe how two groups of patients with COVID-19 who developed a cytokine storm reacted to the two immunomodulators that were added to their treatment, and to compare the clinical and biochemical evolution. The study did not include a comparative analysis of treatment response by patient gender. However, when we analyzed the files by sex, we found that women in both groups had higher values of CRP, LDH and ferritin than men when initiating treatment for the cytokine storm. Moreover, another observation is that the parameters of inflammation (CRP, LDH and ferritin) had a faster improvement in women in both groups than in men. One possible explanation may be that women produce higher levels of proinflammatory cytokines, but women also have an intense and prolonged innate immune response (humoral and cell-mediated), leading to a faster and greater response to viral infections [32].

## 5. Conclusions

Cytokine storms may occur both in moderate and severe forms of COVID-19 pneumonia; in our study, women, especially from the urban environment, were predominantly affected. From comorbidities, we observed that obesity was the most frequent disease in men and women, in moderate and severe forms. The use of immunomodulators in a cytokine storm, on top of complex therapy (antivirals, antibiotics, anticoagulants and corticosteroids) had favorable effects. From a clinical point of view, we observed the remission of fever, cough, tachypnea and an improvement in oxygen saturation. From a biological point of view, in patients treated with Tocilizumab, C-reactive protein, LDH and ferritin had a favorable evolution more rapidly than in patients treated with Anakinra, which hada slower evolution. A possible explanation for why Anakinra is less effective than Tocilizumab could be that IL-1 does not play such a major role in the cascade of the cytokine storm, as with IL-6.

Patients remained in the hospital as long as it was necessary until the values of CRP, LDH and ferritin became normal, and approximately 28% of the patients were discharged with oxygen therapy at home and a rehabilitation program.

Even if vaccination becomes the most important way to prevent the disease, a part of the population still does not have access to vaccines; moreover, the high mutation rate leads to new viral variants. Thus, public health needs to develop effective therapies for COVID-19 (like specific antivirals) and its complications. Many drugs used for COVID-19 and COVID 19 cytokine storms were used for other pathologies, and they were repurposed to treat these urgent situations; none of the anti-cytokine drugs were used against the SARS-CoV-2 virus, so studies showed conflicting results. Understanding the whole factors involved in the cytokine storm leads to better therapies. For example, JAK inhibitors, Baricitinib and Tofacitinib have shown both antiviral and anti-inflammatory effects [31,32].

The medical community needs to gather information in order to draw a clear conclusion.

## Figures and Tables

**Figure 1 jcm-11-02945-f001:**
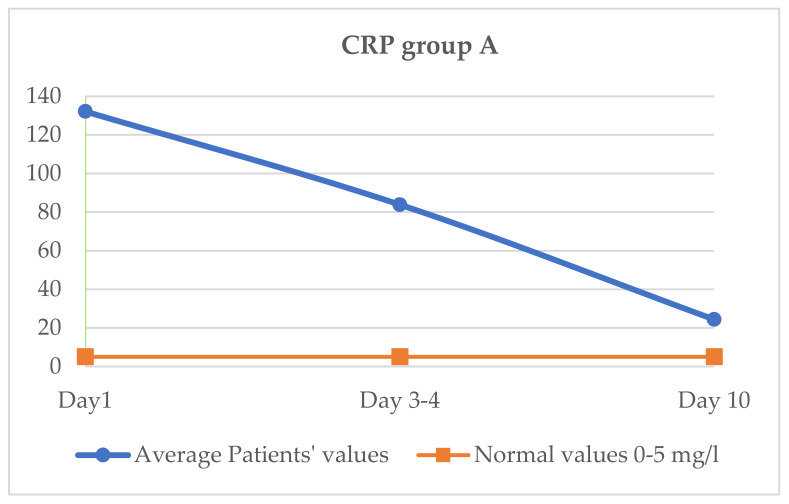
Variation CRP group A.

**Figure 2 jcm-11-02945-f002:**
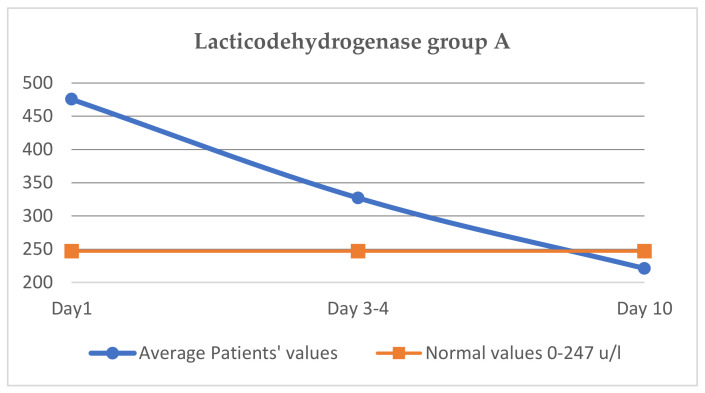
Variation of lactic dehydrogenase group A.

**Figure 3 jcm-11-02945-f003:**
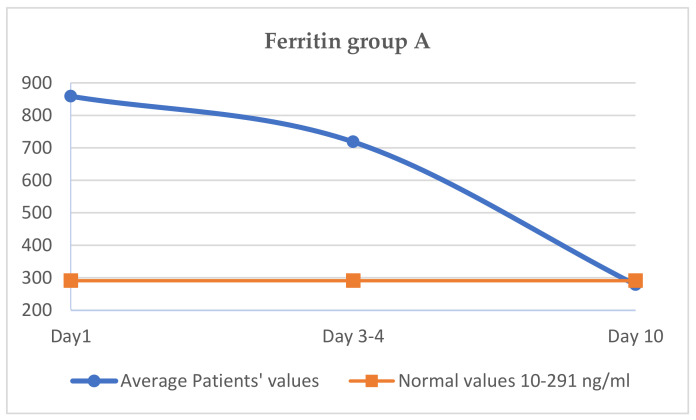
Variation of ferritin group A.

**Figure 4 jcm-11-02945-f004:**
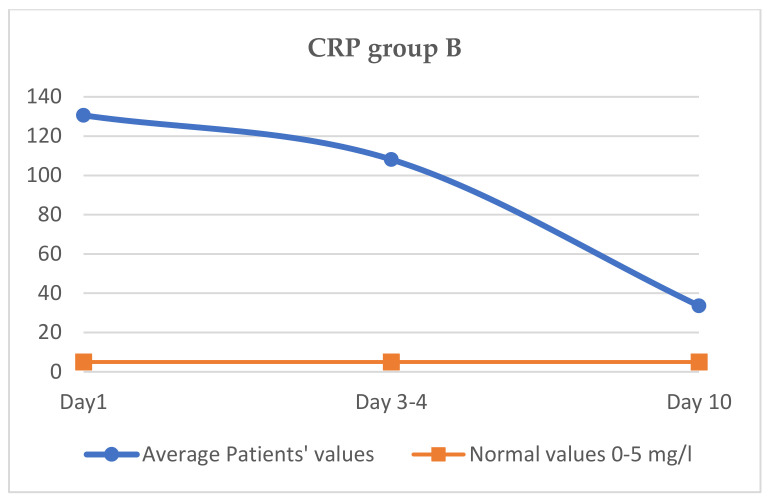
Variation CRP group B.

**Figure 5 jcm-11-02945-f005:**
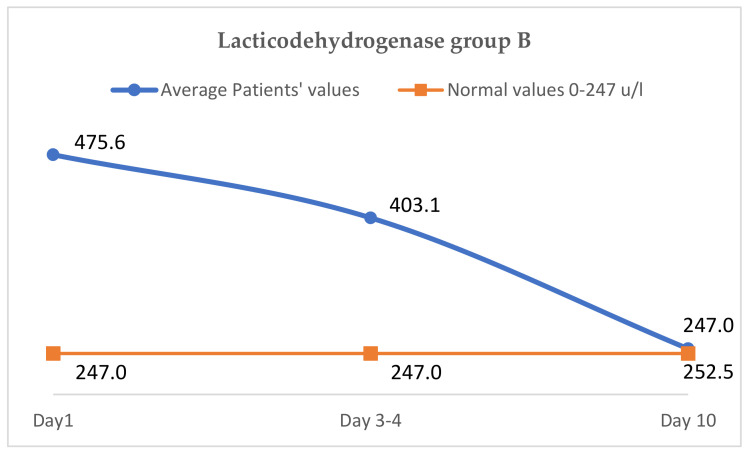
Variation of lactic dehydrogenase group B.

**Figure 6 jcm-11-02945-f006:**
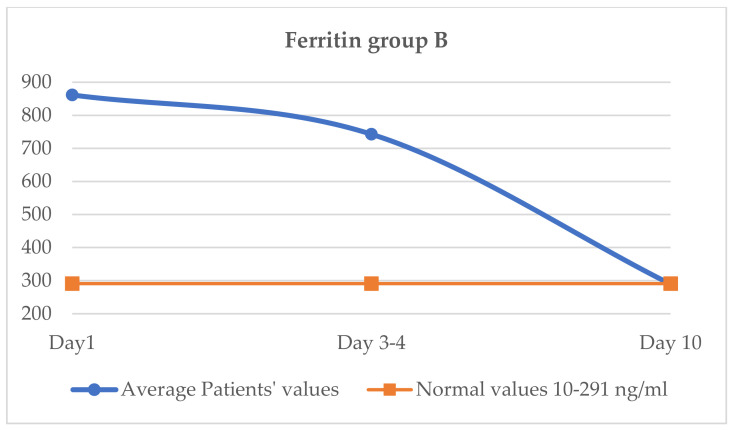
Variation of ferritin group B.

**Table 1 jcm-11-02945-t001:** Characteristics of the population.

Baseline Characteristics of the Group A and Group B	Group A MD ± DS		Group B MD ± DS	
Age (years)	61.18 ± 13.06		62.11 ± 12.02	
Gender Women	94	Percentage% 46.31	109	Percentage% 53.69
Men	55	39.00	86	60.99
Environment Urban	122	62.56	73	37.44
Rural	97	65.10	52	34.90

**Table 2 jcm-11-02945-t002:** Distribution of cases according to the severity of the disease.

Severity of the Disease	Lot A Remdesivir and Tocilizumab		Lot B Remdesivir and Anakinra	
Moderate	36	24.16	10	5.05
Severe	113	75.83	185	94.95

**Table 3 jcm-11-02945-t003:** Distribution of cases by gender.

Gender	Lot A Remdesivir and Tocilizumab		Lot B Remdesivir and Anakinra	
Women	94	39.08%	109	60.92%
Men	55	46.31%	86	52.69%
Total	149		195	

**Table 4 jcm-11-02945-t004:** Distribution of cases by comorbidities.

Comorbidities	Lot A Remdesivir and Tocilizumab		Lot B Remdesivir and Anakinra	
Obesity	51	34%	59	30%
Diabetes mellitus	33	22%	53	27%
Hypertension	24	16%	29	15%
Chronic obstructive lung disease	19	13%	20	10%
Malignancies	9	6%	20	10%
Liver diseases	13	9%	14	8%
Total	149	100%	195	100%

**Table 5 jcm-11-02945-t005:** Evaluation of CRP.

CRP	At Hospitalization	At Day 10 of Administration	P^H−10^	At Discharge
Remdesivir and Tocilizumab (A)	132.58 ± 31.41	24.65 ± 17.30	<0.001	24.02 ± 5.15
Remdesivir and Anakinra (B)	130.11 ± 32.65	40.12 ± 21.57	0.010	41.82 ± 11.62
P^A^^–B^	0.460	0.002		<0.001

P^A–B^—Comparison of *p* values of batch A vs. batch B at Hospitalization and after 10 day; P^H−10^—Comparison of *p* values of batch A vs. batch B at hospitalization and discharge.

**Table 6 jcm-11-02945-t006:** Evaluation of LDH.

LDH	At Hospitalization	At Day 10 of Administration	P^H−10^	At Discharge
Remdesivir and Tocilizumab	475.67 ±11.31	221.65 ± 14.20	<0.001	220.62 ± 5.15
Remdesivir and Anakinra.	475.41 ± 12.65	252.72 ± 27.37	0.011	254.82 ± 11.62
P^A^^–B^	0.340	0.0015		<0.001

P^A–B^—Comparison of *p* values of batch A vs. batch B at Hospitalization and after 10 day; P^H−10^—Comparison of *p* values of batch A vs. batch B at hospitalization and discharge.

**Table 7 jcm-11-02945-t007:** The evaluation of ferritin.

Ferritin	At Hospitalization	At Day 10 of Administration	P^H−10^	At Discharge
Remdesivir and Tocilizumab	859.67 ±21.31	302.65 ± 27.30	<0.00	289.62 ± 5.15
Remdesivir and Anakinra.	861.41 ± 31.65	318.72 ± 21.47	0.01	291.82 ± 11.62
P^A^^–B^	0.010	0.002		<0.001

LDH—Lactate dehydrogenase; P^H−10^—Comparison of *p* values of batch A vs. batch B at hospitalization and discharge; P^A–B^—Comparison of *p* values of batch A vs. batch B at Hospitalization and after 10 day.

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
