# Peer review of "Comparative Study of Cytokine Storm Treatment in Patients with COVID-19 Pneumonia Using Immunomodulators"

_jcm, 2022, doi:10.3390/jcm11102945_

Round 1
Reviewer 1 Report
- Summary and abstract need to be improved.
- SARS-CoV-2 is the correct way of writing, edit all incorrect words.
- Line 39-40 should be rewritten
- what is the meaning of lesion in line 43?
- line 52: add 'and' before renal impairment
- line 54 isnt clear, biochemistry?
- line 61: its cardiomyopathy
- line 62: studies show
- line 74 is not clear
- line 77 : why is this sign used = ?
- line 81 should be corrected
- Table 1 isnt clear
- The study must segregate their data according to the sex of patients. Males and females have different pharmacodynamics as well as pharmacokinetics with respect to medications.So the data should be separately addressed.
- The data on comorbidity predisposition of patients should be tabulated.
Author Response
The authors acknowledge the useful observations and suggestions of the reviewer’s as concerns the manuscript entitled: Comparative study of cytokine storm treatment in patients with Covid 19 pneumonia using immunomodulators co-authored by Felicia Marc, Corina Maria Moldovan, Anica Hoza, Sorina Magheru, Gabriela Ciavoi, Dorina Maria Farcas, Liliana Sachelarie*, Gabriela Calin, Laura Romila*, Daniela Damir*,Alexandru Gratian Naum
According to the reviewer’s recommendations, all the suggestions were taken into account, as follows:
- Summary and abstract need to be improved.
R: Infection with SARS-CoV-2 virus may involve many organs and systems, predominantly the lungs. Patients with viral pneumonia, moderate or severe, are at risk to develop cytokine storm, which consists of a clinical and paraclinical deterioration of the disease,usually around days 8-12 from the onset. For the therapy of cytokine storm, immunomodulation has been shown to be very important, since it was proven that an immune disregulation occurs.In our study, we compared the efectiveness of two interleukin antagonists (IL-6 antagonist, Tocilizumab and IL-1 antagonist, Anakinra) used on top of the treatment for cytokine storm, in 2 groups of patients. Both groups had a clinical and biological improvement; in Tocilizumab group the beneficial effect was observed faster, while in Anakinra group, the effect occurred slower. The findings bring additional information from a clinical point of view, showing that these two immunomodulators are beneficial.
Abstract: (1) Background. In patients hospitalized with COVID-19 pneumonia, especially moderate and severe forms, cytokine storm may occurr, characterized by worsening of symptoms and alteration of biological parameters on days 8-12 of the disease. Therapeutic options for cytokine storm are still controversial, requiring further clarification; (2) Methods: Our study included 344 patients with moderate and severe pneumonia admitted to the internal medicine department who developed cytokine storm (diagnosed by clinical and biochemical criteria). In group A, 149 patients were treated with Remdesivir and Tocilizumab (together with other drugs, including corticosteroids, antibiotics, anticoagulants), and in group B, 195 patients received Remdesivir and Anakinra. Patients were monitored clinically and by laboratory tests, the main biochemical parameters being CRP (C-reactive protein), LDH (lactic dehydrogenase) and ferritin; (3) Results: Patients were followed up from a clinical point of view and also by measurement of CRP, LDH and ferritin at the beginning of therapy, on days 3-4 and on the tenth day. In both groups we registered a clinical improvement and a decrease of the parameters of cytokine storm. In group A, with IL-6 antagonist, Tocilizumab, the beneficial effect occured faster; in group B, with IL-1 antagonist, Anakinra, the beneficial effect was slower . (4) Conclusions: The use of immunomodulators in the cytokine storm, Tocilizumab and Anakinra, showed favourable effects, both clinical and biochemical.
- SARS-CoV-2 is the correct way of writing, edit all incorrect words.
R: I edited all the incorrect words.
- Line 39-40 should be rewritten
R: The most commonly affected organ in this disease is the lung - thus, virus may cause viral pneumonia, which is classified after clinical and paraclinical evaluation in mild, moderate and severe. Usually, a superimposed bacterial infection occurs, so pneumonia becomes viral and bacterial. [1,2,3 ,4).In addition to lung damage (cough, shortness of breath, chest pain, haemoptysis, wheezing, rhinorrhea, odynophagia, nasal congestion, anosmia, ageuzia), SARS-CoV-2 virus may cause other organs and systems involvement, such as: general manifestations....
- line 52: add 'and' before renal impairment
R: I added
- line 54 isnt clear, biochemistry?
R: In COVID-19 pneumonia, on days 8-12 of the disease, some patients develop the "cytokine storm". This is characterised by an aggravation of the clinical status (fever, chills, dyspnea,tachypnea, a decrease in oxygen saturation that requires oxygen supplementation) and by alteration of blood tests [8,9,10]. The biochemical criteria for the detection...
- line 61: its cardiomyopathy
R: I replaced
- line 62: studies shown
R: I replaced
- line 74 is not clear
R: The classic treatment plan, according to the Romanian Ministry of Health recommendations protocol, includes an antiviral (Favipiravir or Remdesivir), anticoagulant, corticosteroids (Dexamethasone- which acts as anti-inflammatory and as immunomodulator), antibiotic, symptomatic ( Acetaminophen for fever, Codein phosphate or Acetilcisteine for cough, Metoclopramid for vomiting, etc)[16,17,18].
- line 77 : why is this sign used = ?
R: I fixed it
- line 81 should be corrected
R: I corrected
- Table 1 isnt clear
R:
- Characteristics of the population (sex, age and environment)
Tabel 1. Characteristics of the population
|
|
No. |
Percentage% |
Min/Max |
MD±DS |
|
Gender (F/M) |
203 women 141 men |
59.10% 40.90% |
|
|
|
Age (years) |
|
|
26-88 |
61,18±13,06 |
|
Environment (U/R) |
195 urban 149 rural |
56.18 % 43.82% |
|
|
In our study group over 50% were women (59.10%), the ratio of women / men was 1: 1, the age was between 26-88 years, the average age was 60.18 years, and the patients came mainly from the urban environment (56.18%), the urban / rural ratio being1.3: 1.
The study must segregate their data according to the sex of patients. Males and females have different pharmacodynamics as well as pharmacokinetics with respect to medications.So the data should be separately addressed.
- The data on comorbidity predisposition of patients should be tabulated
Table 4 Distribution of cases by comorbidities
|
Comorbidities |
Lot A Remdesivir and Tocilizumab |
Lot B Remdesivir and Anakinra |
||
|
Obesity |
51 |
34% |
59 |
30% |
|
Diabetes mellitus |
33 |
22% |
53 |
27% |
|
Hypertension |
24 |
16% |
29 |
15% |
|
Chronic obstructive lung disease |
19 |
13% |
20 |
10% |
|
Malignancies |
9 |
6% |
20 |
10% |
|
Liver diseases |
13 |
9% |
14 |
8% |
|
Total |
149 |
100% |
195 |
100% |
I remain most respectfully yours,
Prof.dr. Liliana Sachelarie

Reviewer 2 Report
The present study is interesting and plays an important role in the development of therapeutic regimen against COVID-19. However, address the following points to improve the manuscript.
- Replace Sars-Cov 2 to SARS-CoV-2 throughout the manuscript
- Replace Covid-19 to COVID-19 throughout the manuscript
- Line no.53 to 55: is not clear, reframe the sentences for clarity and explain what do you mean by biochemistry.
“In Covid-19 pneumonia, on days 8-12 of the disease, some patients have a clinical aggra-53 vation (fever, severe dyspnea, tachypnea, decreased oxygen saturation) and biochemistry 54 (cytokine storm) [8,9,10].”
- From Line no. 62 to 66: Provide the following references
https://doi.org/10.3390/vaccines9050436
- Line No. 72: Ministry of Health recommendations 72 protocol, [Mention the country].
- Line no. 74: what does symptomatic mean?
- Line No. 175: Evolutin?? Change into Evaluation.
- Check throughout the manuscript for the “Evolution”
Evaluation is the appropriate word.
- Figure legends: All the legends of each figure should be descriptive and explanatory for better readability.
- Table 3. Evolution of CRP? Replace the word Evolution with Evaluation
- Table no.4: please justify why the Levels of LDH increased in the Remdesivir and Anakinra at the time of discharge.
- Discussion:
Explain the advantages of Tocilizumab over the other anti-inflammatory drugs, add few lines from the following article. Explain the crucial role of IL-6 in the imitation of cytokine storm which can be inhibited with IL-6 inhibitor such as Tocilizumab. Incorporate information from the following articles.
Tocilizumab and COVID-19: Timing of Administration and Efficacy
https://doi.org/10.3389/fphar.2022.825749
https://doi.org/10.1080/21645515.2020.1822137
Explain the Why Anakinra is lees effective as compared to Tocilizumab.
- Improve the conclusion by providing the major findings of the research. Additionally, provide the future directions to develop more effective therapeutic drugs to treat COVID-19.
- Explain how these therapeutic options will be reliable alternatives amid the emergence of novel variants of SARS-CoV-2.
- Importantly, the language needs to improve for clear information. This will increase the readability.
Author Response
The authors acknowledge the useful observations and suggestions of the reviewer’s as concerns the manuscript entitled: Comparative study of cytokine storm treatment in patients with Covid 19 pneumonia using immunomodulators co-authored by Felicia Marc, Corina Maria Moldovan, Anica Hoza, Sorina Magheru, Gabriela Ciavoi, Dorina Maria Farcas, Liliana Sachelarie*, Gabriela Calin, Laura Romila*, Daniela Damir*,Alexandru Gratian Naum
According to the reviewer’s recommendations, all the suggestions were taken into account, as follows:
The present study is interesting and plays an important role in the development of therapeutic regimen against COVID-19. However, address the following points to improve the manuscript.
- Replace Sars-Cov 2 to SARS-CoV-2 throughout the manuscript - I replaced Sars-Cov 2 to SARS-CoV-2 throughout the manuscript
- Replace Covid-19 to COVID-19 throughout the manuscript -I replaced Covid-19 to COVID-19 throughout the manuscript
- Line no.53 to 55: is not clear, reframe the sentences for clarity and explain what do you mean by biochemistry.
In COVID-19 pneumonia, on days 8-12 of the disease, some patients develop the "cytokine storm". This is characterised by an aggravation of the clinical status (fever, chills, dyspnea,tachypnea, a decrease in oxygen saturation that requires oxygen supplementation) and by alteration of blood tests [8,9,10].
The biochemical criteria for the detection of cytokine storm are: elevation of C-reactive protein (CRP) over 30 mg / l (normal value 0-5 mg / l), increase of lactate dehydrogenase (LDH) over 300 U / L (normal value 0-247 U / L) and increase of ferritin over 700 ng / ml (normal value 10-291 ng / ml) [11,12].
“In Covid-19 pneumonia, on days 8-12 of the disease, some patients have a clinical aggravation (fever, severe dyspnea, tachypnea, decreased oxygen saturation) and biochemistry (cytokine storm) [8,9,10].”
- From Line no. 62 to 66: Provide the following references
https://doi.org/10.3390/vaccines9050436
Studies show that COVID- 19 patients have increased levels of numerous inflammatory cytokines,like IL-1 beta, IL-2, IL-6,TNF-alpha,etc and these cytokines correlate with disease severity. The release of numerous cytokines is associated with the clinical manifestations like headache, chills, diziness, fatigue, fever, lung damage, vascular leaking, heart failure, synthesis of acute-phase proteins [13].
- Line No. 72: Ministry of Health recommendations 72 protocol, [Mention the country].
The classic treatment plan, according to the Romanian Ministry of Health recommendations protocol, includes an antiviral (Favipiravir or Remdesivir), anticoagulant, corticosteroids (Dexamethasone- which acts as anti-inflammatory and as immunomodulator), antibiotic, symptomatic ( Acetaminophen for fever, Codein phosphate or Acetilcisteine for cough, Metoclopramid for vomiting, etc)[16,17,18].
- Line no. 74: what does symptomatic mean?
symptomatic ( Acetaminophen for fever, Codein phosphate or Acetilcisteine for cough, Metoclopramid for vomiting, etc)
- Line No. 175: Evolutin?? Change into Evaluation. - I changed
- Check throughout the manuscript for the “Evolution” - I checked
Evaluation is the appropriate word. - I changed
- Figure legends: All the legends of each figure should be descriptive and explanatory for better readability.- Done
- Table 3. Evolution of CRP? Replace the word Evolution with Evaluation - Done
- Table no.4: please justify why the Levels of LDH increased in the Remdesivir and Anakinra at the time of discharge.- Done
- Discussion:
Explain the advantages of Tocilizumab over the other anti-inflammatory drugs, add few lines from the following article. Explain the crucial role of IL-6 in the imitation of cytokine storm which can be inhibited with IL-6 inhibitor such as Tocilizumab. Incorporate information from the following articles.
The hyperinflammatory status which occurs in cytokine storm is caused by the release of numerous proinflammatory cytokines, such as IL-6, IL-1, TNF-alpha,etc., with IL-6 as the orchestral lead for COVID-19 cytokine release syndrome (32,33). The secretion of IL-6 causes vascular leak syndrome, triggering coagulation and complement pathways; also, IL-6 plays a crucial role to induce a panoply of other proinflammatory cytokines and chemokines, continuing the process of inflammation and thrombosis that can finally produce multiple organ failure.(32,33).
Tocilizumab is currently standing out as an IL-6 receptor blockade that might interrupt the inflammatory cascade at a crucial stage. Results from REMAP-CAP study suggest that Tocilizumab is most effective when administered early (within 24 hours of organ failure), in patients with progressive disease and substantial oxygen requirements.
The primary laborator indicator that determines the proper time to start Tocilizumab should be the blood IL-6 level; similarly, in order to start Anakinra, the levels of IL-1 should be determined. Unfortunately, these tests were not available for us
Explain the Why Anakinra is lees effective as compared to Tocilizumab.
- Improve the conclusion by providing the major findings of the research. Additionally, provide the future directions to develop more effective therapeutic drugs to treat COVID-19.
- Explain how these therapeutic options will be reliable alternatives amid the emergence of novel variants of SARS-CoV-2.
- Importantly, the language needs to improve for clear information. This will increase the readability.
Cytokine storm may occur both in moderate and severe forms of COVID-19 pneumonia; in our study, women, especially from urban envinronment, were predominantly affected. From the comorbidities, we observed that obesity was the most frequent disease in men and women, in moderate and severe forms. The use of immunomodulators in cytokine storm, on top of a complex therapy (antivirals, antibiotics, anticoagulants, corticosteroids) had favourable effects. From a clinical point of view, we observed remission of fever, cough, tachypnea, an improvement of oxygen saturation. From a biological point of view, in patients treated with Tocilizumab, C reactive protein, LDH and ferritin had a favourable evolution in a more rapid way than in patients treated with Anakinra, with a slower evolution. A possible explanation why Anakinra is less effective than Tocilizumab could be that IL-1 doesn't play such a major role in the cascade of cytokine storm as IL-6.
Patients remained in hospital as long as it was necesary, until the values of CRP, LDH and ferritin became normal and approximately 28% of the patients were discharged with oxygenotherapy at home and rehabilitation programme.
Even if vaccination becomes the most important way to prevent the disease, a part of the population still doesn't have access to vaccine; also, the high mutation rate leads to new viral variants. Thus, it is very important for the public health to develop effective therapies for COVID- 19 (like specific antivirals) and its complications. Many drugs used for COVID- 19 and for COVID 19 cytokine storm were used for other pathologies and they were repurposed in order to treat these urgent situations; none of the anti-cytokine drugs were used against SARS-CoV-2 virus, so studies showed conflicting results. Understanding the whole factors involved in cytokine storm leads to better therapies- for example, JAK inhibitors, Baricitinib and Tofacitinib have shown both antiviral and antiinflammatory effects.
I remain most respectfully yours,
Prof.dr. Liliana Sachelarie

Round 2
Reviewer 1 Report
The effect of medicine combination on female cohort is still missing from the manuscript. There has to be clear demarcation between the response of drug combinations in female vis-a-vis male patients.
Author Response
The authors acknowledge the useful observations and suggestions of the reviewer’s as concerns the manuscript entitled: Comparative study of cytokine storm treatment in patients with Covid 19 pneumonia using immunomodulators by
Felicia Marc, Corina Maria Moldovan, Anica Hoza, Sorina Magheru, Gabriela Ciavoi, Dorina Maria Farcas, Liliana Sachelarie, Gabriela Calin, Laura Romila, Daniela Damir,Alexandru Gratian Naum
According to the reviewer’s recommendations, all the suggestions were taken into account, as follows:
The effect of medicine combination on female cohort is still missing from the manuscript. There has to be clear demarcation between the response of drug combinations in female vis-a-vis male patients.
R:
The aim of our study was to observe how 2 groups of patients with COVID-19 who developed a cytokine storm, reacted to the 2 immunomodulators that were added to their treatment and to compare the clinical and biochemical evolution. The study did not include a comparative analysis of treatment response by patient gender. However, when we analyzed the files by sex, we found that women in both groups had higher values of CRP, LDH and ferritin than men when initiating treatment for the cytokine storm. Also, another observation is that the parameters of inflammation (CRP, LDH, ferritin) had a faster improvement in women in both groups than in men. One possible explanation may be that women produce higher levels of proinflammatory cytokines, but women also have an intense and prolonged innate immune response (humoral and cell-mediated), leading to a faster and greater response to viral infections. [Annalisa C.; Francesco R.; Giuseppe P. Covid-19 Kills More Men Than Women: An Overview of Possible Reasons. Front Cardiovasc Med. 2020; 7: 131].
Thank you very much for review reports and for the extremely useful observations and suggestions!
Kind regards,
Prof.dr. Liliana Sachelarie

Reviewer 2 Report
The manuscript has been revised sufficiently with appropriate references. However, there are several languages errors and grammatical errors. I recommend that the article should be critical investigated for such issues.
Additionally, I recommend that the professional English editing services can be considered to improve the readability.
Best Regards
Author Response
The authors acknowledge the useful observations and suggestions of the reviewer’s as concerns the manuscript entitled: Comparative study of cytokine storm treatment in patients with Covid 19 pneumonia using immunomodulators by
Felicia Marc, Corina Maria Moldovan, Anica Hoza, Sorina Magheru, Gabriela Ciavoi, Dorina Maria Farcas, Liliana Sachelarie, Gabriela Calin, Laura Romila, Daniela Damir,Alexandru Gratian Naum
According to the reviewer’s recommendations, all the suggestions were taken into account, as follows:
The manuscript has been revised sufficiently with appropriate references. However, there are several languages errors and grammatical errors. I recommend that the article should be critical investigated for such issues.
Additionally, I recommend that the professional English editing services can be considered to improve the readability.
R:
We checked the article from the point of view of languages errors and grammatical errors.
Thank you very much for review reports and for the extremely useful observations and suggestions!
Kind regards,
Prof.dr. Liliana Sachelarie
